# Non-canonical genomic driver mutations of urethane carcinogenesis

Siqi Li[¤], Christopher M. Counter [ID]*

Department of Pharmacology & Cancer Biology, Duke University, Durham, NC, United States of America

¤ Current address: Human Biology Division, Fred Hutchinson Cancer Research Center, Seattle, WA, United States of America
* count004@mc.duke.edu

**Data Availability Statement:** The raw sequencing data for WES datasets was deposited to NCBI Sequence Read Archive (SRA) under accession number PRJNA663179. MDS datasets were previously generated [7,11] and the raw sequencing data was deposited to NCBI Sequence

## Abstract

The carcinogen urethane induces pulmonary tumors in mice initiated by an incredibly specific $Q_{61}L/R$ oncogenic mutation in the proto-oncogene *Kras*. Previous Whole-Exome Sequencing of urethane-induced tumors revealed a bias towards A→T/G and G→A substitutions. Subsequent ultra-sensitive Maximum-Depth Sequencing of *Kras* shortly after urethane exposure suggest a further refinement to **C$\underline{\textbf{A}}$→C$\underline{\textbf{T/G}}$** substitutions. As $C_{182}\underline{\textbf{A}}A$→$C_{182}\underline{\textbf{T/G}}A$ substitutions in *Kras* result in $Q_{61}L/R$ mutations, the extreme bias of urethane towards these genomic driver mutations can be ascribed to the specificity of the carcinogen for C$\underline{\textbf{A}}$→C$\underline{\textbf{T/G}}$ substitutions. However, we previously found that changing rare codons to common in the *Kras* gene to increase protein expression shifted mutations in urethane-induced tumors away from *Kras*, or when detected in *Kras*, to G12D mutations that are usually rarely detected in such tumors. Moreover, the loss of p53 partially reversed this effect, generating tumors with either Q61L/R or G12D oncogenic *Kras* mutations, or no *Kras* mutations, presumably due to other genomic driver mutations. Determining the origin of these G12D and other unknown non-canonical genomic driver mutations would provide critical insight into the extreme bias of carcinogens for specific genomic driver mutations. We thus compared the types of Single Nucleotide Variations detected by previously performed Maximum-Depth Sequencing immediately after urethane exposure to the mutation signatures derived from Whole Exome Sequencing of urethane-induced tumors. This identified two types of non-canonical mutations. First, a $V_{637}E$ oncogenic mutation in the proto-oncogene *Braf* that conforms to the mutation signature of urethane, suggesting that the mutational bias of the carcinogen may account for this non-canonical mutation, similar to that for canonical Q61L/R mutations in *Kras*. Second, $G_{12}D$ and $Q_{61}H$ mutations in *Kras* that did not fit this mutation signature, and instead shared similarity with Single Nucleotide Variations detected by Maximum-Depth Sequencing from normal cells, suggesting that perhaps these mutations were pre-existing. We thus posit that when canonical *Kras* mutations are selected against that the carcinogen may instead promote the expansion of pre-existing genomic driver mutations, although admittedly we cannot rule out other mechanisms. Interrogating the mutation signatures of human lung cancers similarly identified *KRAS* genomic driver mutations that failed to match the mutation signature of the tumor. Thus, we also speculate that the selection for non-canonical genomic driver mutations during urethane

Read Archive (SRA) under accession number PRJNA561927 and PRJNA663179.

**Funding:** This work was supported by the National Cancer Institute (USA) grants R01CA123031 and P01CA203657 as well as an internal voucher grant from the Duke University School of Medicine to CMC. The Duke Cancer Institute Sequencing and Genomic Technologies Shared Resource that performed sequencing for this study is supported by P30CA014236. There was no additional external funding received for this study.

**Competing interests:** CMC is co-Founder of Merlon Inc, a member of the External Advisory Board for the University of Colorado Cancer Center, has a cross appointment with Duke-NUS, is an ex officio of the executive team for the Cancer Biology Training Consortium, has previously consulted for the Guidepoint Network in an ad hoc fashion and received licensing reimbursement from Humacyte Inc in the past. These relationships did not provide any salary or research support, did not play any role in the study design, data collection and analysis, decision to publish, and do not alter our adherence to PLOS ONE policies on sharing data and materials.

carcinogenesis may reflect the process by which discordance between genomic driver mutations and mutational signatures arises in human cancers.

## Introduction

Intraperitoneal injections of urethane, a carcinogen detected in fermented and alcoholic products [1], induces pulmonary tumors in mice characterized by either a $Kras^{Q61L}$ or $Kras^{Q61R}$ genomic driver mutation, depending on the mouse strain [2]. Urethane is known to induce ethenodeoxyadenosine adducts [3, 4]. Consistent with this type of DNA damage, _W_hole _E_xome _S_equencing (WES) of urethane-induced tumors revealed a bias towards A➛T transversions as well as A➛G and G➛A transitions [5]. Capitalizing upon the development of the ultra-sensitive _M_aximum _D_epth _S_equencing (MDS) to detect _de novo_ mutations in bacteria [6], we previously adapted this assay to screen for _S_ingle _N_ucleotide _V_arious (SNV) in the _Kras_ gene of murine lungs shortly after the animals were exposed to urethane. This analysis revealed a more restricted mutation signature of a C followed by an A➛T transversion, and to a lesser extent, an A➛G transition [7]. Importantly, C_**A**_➛C_**T/G**_ substitutions convert codon $C_{182}AA$ encoding $Q_{61}$ in _Kras_ to the $C_{182}\underline{\textbf{T}}A$ ($Q_{61}L$) and proportionally the less common $C_{182}\underline{\textbf{G}}A$ ($Q_{61}R$) oncogenic mutations found in urethane-induced tumors of the A/J strain of mice. Moreover, previous WES analysis of urethane-induced tumors failed to identify other highly recurrent co-operating mutations, arguing that $Kras^{Q61L/R}$ mutations are indeed the genomic driver mutations in urethane carcinogenesis [7]. Collectively, these data support a model whereby the mutational preference of this carcinogen underlies the extreme bias for the $Kras^{Q61L/R}$ genomic driver mutations characteristic of urethane-induced tumors [5, 7, 8].

We previously reported that the _Kras_ gene of mice (and humans) is enriched in rare codons, and further, that this bias towards rare codons reduces protein levels [9]. We further found that converting 27 rare codons to their common counterparts in the third coding exon of the endogenous _Kras_ murine gene (termed $Kras^{ex3op}$), which does not contain the sites for oncogenic mutations, not only reduced the number of urethane-induced tumors with a _Kras_ genomic driver mutation, but in those tumors in which _Kras_ was mutated, a completely different oncogenic mutation ($G_{12}D$) was recovered [10, 11]. We suggest that these latter mutations arose due to selection of the less active $G_{12}D$ mutation in the more highly expressed $Kras^{ex3op}$ allele to avoid oncogenic stress. Indeed, high oncogenic RAS expression is known to induce senescence [12], a $G_{12}D$ mutation exhibits lower levels of both active GTP-bound Kras [10, 13] and activation of downstream target genes [11] compared to a $Q_{61}R$ mutation, and loss of the tumor suppressor _Trp53_ to suppress oncogenic stress generates urethane-induced tumors with a $Q_{61}R$ (in addition to $G_{12}D$) mutation in the $Kras^{ex3op}$ allele [11]. Of special note, a $G_{12}D$ mutation arises from a $G\underline{\textbf{G}}_{35}T$➛$G\underline{\textbf{A}}T$ transversion that does _not_ match the aforementioned C_**A**_➛C_**T/G**_ mutation signature of urethane-induced tumors.

Given that these mutations were discordant with the mutation signature of urethane, and that some tumors arose without an oncogenic mutation in _Kras_, we sought to identify the types of non-canonical mutations induced by urethane and then their potential origins, to elucidate how the extreme bias of urethane for $Kras^{Q61L/R}$ mutations is reprogrammed. We thus compared the SNV previously detected by MDS in the _Kras_ gene shortly after urethane exposure against the mutation signature derived by WES analysis of urethane-induced tumors. This identified two types of non-canonical genomic driver mutations. First, genomic driver mutations that tracked with the mutation signature of urethane, suggesting that the mutagenic

activity of the carcinogen induced these mutations. Second, genomic driver mutations that tracked with SNVs detected in normal cells, speculatively suggesting that perhaps the carcinogen may instead have acted to promote a pre-existing driver mutation.

# Materials and methods

## Generation of WES datasets

Urethane induced tumors were isolated at the time of necropsy from 23 urethane-induced lung tumors from $Sftpc^{CreER/CreER};Trp53^{fl/fl}$ mice [11] injected with tamoxifen with none (6 tumors), one (10 tumors), or two (7 tumors) $Kras^{ex3op}$ alleles (**S1 Table**). As previously described [11], mice (6–8 weeks old, male and female) were injected intraperitoneally with tamoxifen (Sigma T5648) dissolved in corn oil at a dose of 0.25 mg/g body weight every other day for a total of four doses. One week later all mice were injected intraperitoneally once with urethane (Sigma U2500) dissolved in sterile PBS at a dose of 1 mg/g body weight. Animals were maintained under pathogen-free conditions and visually monitored and weighed weekly. Approximately 12 months after urethane injection, mice were humanely euthanized by $CO_2$ asphyxiation followed by cervical dislocation, after which the lung tumors were harvested. Genomic DNA was isolated from the tumors as previously described [11]. 50 μg of this DNA was sheared to ~400 bp fragments using the Covaris E210 system. Libraries were generated with the Twist Mouse Exome Panel (Twist Bioscience 101242), hybridized, and enriched according to the manufacturer's instructions. Libraries were pooled and sequenced through 150 bp paired-end reads on an Illumina NovaSeq 6000 S-Prime flow cell. Analysis of WES data was performed on the Galaxy platform [14]. Sequencing adaptor was trimmed from the raw sequencing reads using Trimmomatic [15]. Trimmed reads were aligned to GRCm38/mm10 version of the *Mus musculus* genome using BWA-MEM [16]. Successfully mapped reads with a minimal mapping quality of 1, and for which the mate read has also been mapped were selected using the tool 'Filter BAM datasets on a variety of attributes'[17]. PCR duplicates were removed using RmDup [18]. Genomic regions targeted by the Twist Mouse Exome Panel were selected using the tool 'Slice BAM by genomic regions' [19] and bed dataset 'Twist_Mouse_Exome_Target_Rev1_7APR20.bed' from Twist Bioscience website (https://www.twistbioscience.com/resources/bed-file/twist-mouse-exome-panel-bed-file). Pileup file was generated from the BAM dataset using the tool 'Generate pileup from BAM dataset' [20]. On average, samples were sequenced with a sequencing depth of 90× with 74% of the exome covered by >20x. The raw sequencing data for WES datasets was deposited to NCBI Sequence Read Archive (SRA) under accession number PRJNA663179. Animal studies were approved in writing by the Duke University Institutional Animal Care Committee.

## Generation of MDS datasets

MDS datasets were previously generated [7, 11] and the raw sequencing data was deposited to NCBI Sequence Read Archive (SRA) under accession number PRJNA561927 and PRJNA663179.

## Variant identification, prioritization, and validation

Copy number alterations were identified using CNVkit-v0.9.10 [21]. SNVs were identified using VarScan [22]. GRCm38/mm10 served as the reference during calling and the minimum read depth for variants was set at 8. Called variants were annotated using SnpEff eff [23]. SNVs with variant allele frequency (VAF) < 5% were discarded. Variant calls from different samples were combined into a merged set using 'bcftools merge' [20]. The following SNVs were

considered as SNPs and removed: SNPs reported by the Mouse Genome Project of the Sanger Institute in the 129S1_SvImJ strain background [24]; SNVs appearing together in the same group of samples; SNVs clustering together by genomic coordinates in individual samples. High-likelihood cancer genomic driver genes were initially prioritized from a list of 460 previously defined cancer genes [25]. To focus on clonal genomic driver mutations, non-synonymous SNVs in the selected cancer genes were further filtered by VAF higher than the mean-3SD of the VAFs of *Kras* mutations determined from Kras-mutant tumors (15.7%). SNVs with VAF of 100% were considered as a SNP and removed.

To expand the list of putative genomic driver mutations, non-synonymous SNVs in all genes were filtered by VAF between the mean-3SD and mean+3SD of the VAFs of *Kras* mutations determined from Kras mutant tumors (15.7%-72.2%). SNVs clustering in the same gene in the same sample were removed. SNVs were further limited to genes for which the human homolog has single non-synonymous SNV occurring in at least five cancer patients in the ICGC data portal [26]. SNVs with VAF outside the 150% or 50% range of the VAF of Kras or Braf$^{V637E}$ mutations in Kras- or Braf$^{V637E}$-mutant samples were removed. SNVs occurring in more than eight samples were considered as SNPs and removed. SNVs were further limited to amino acid positions that are conserved and mutated in human and/or mutated to the same amino acid in human to shorten the list of putative genomic driver mutations for each tumor to ~5. This list of SNVs were compiled with the SNVs selected from defined cancer genes above that occur at amino acid positions mutated in cancer patients. SNVs with VAF < 65% of the VAF of Kras or Braf$^{V637E}$ mutations in Kras- or Braf$^{V637E}$-mutant tumors, or the putative genomic driver mutation with the highest VAF in the other tumors, were removed.

## Validation of SNVs

SNVs were validated by Sanger sequencing of the PCR product from either genomic DNA or cDNA. SNVs were called validated if they were detected in the tumor but not in matched normal lung tissue. Sanger validation was prioritized for mutations recurrent across tumors as well as mutations with the highest VAF in each tumor. The primers used were listed in **S4 Table** and validation results were shown in **S2 Table**.

## Comparison of the mutation spectrum of MDS and WES

For the mutation spectrum of MDS, substitutions detected in *Kras* exon 1 and exon 2, *Hras* exon 2 in the lungs of A/J strain of mice one week after urethane or PBS exposure in a previous study [7] and well as substitutions detected in *Kras* exon 1 and exon 2 in the lungs of 129 strain of mice one week after urethane or PBS exposure in another study [11], were compiled together (**S3 Table**). A previously published study has shown that the C➔T or G➔T substitutions from the strand sequenced by MDS assay has a high mutation frequency not reflected in the complementary strand [7], thus those substitutions were considered as potential technical artifacts and removed from further analysis. Substitutions with frequency higher than $2 \times 10^{-5}$ were considered as outliers and removed. Substitutions were annotated by the 96 possible trinucleotide context substitutions (6 types of substitutions × 4 possible flanking 5' bases × 4 possible flanking 3' bases) and the mutation frequency for each one of the 96 substitutions averaged across all nucleotide positions in all mice were plotted. The average mutation frequency calculated PBS-treated mice were multiplied by $10^6$ and used as mutagenic optimality score for PBS in **Figs 1A** and **3B**. The average mutation frequency of urethane-treated mice was normalized to PBS-treated mice by deducting the average mutation frequency of PBS-treated mice from that of urethane-treated mice for each of the 96 substitutions. This normalized mutation frequency was multiplied by $10^6$ and used as mutagenic optimality score for

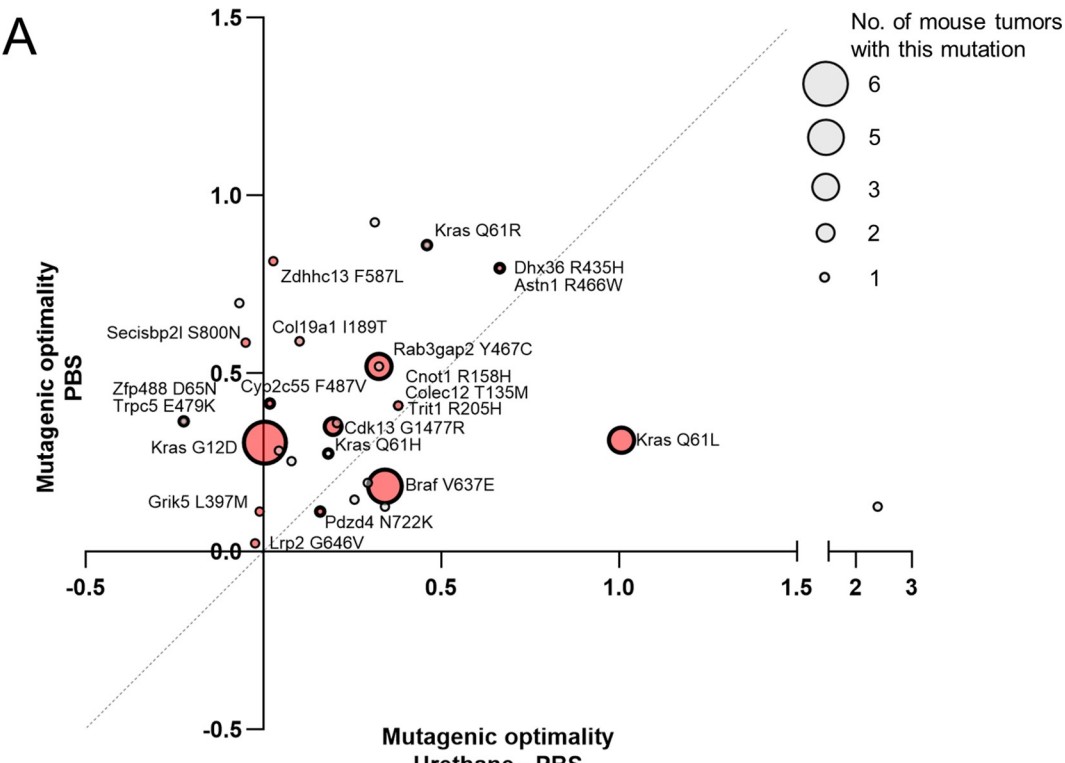

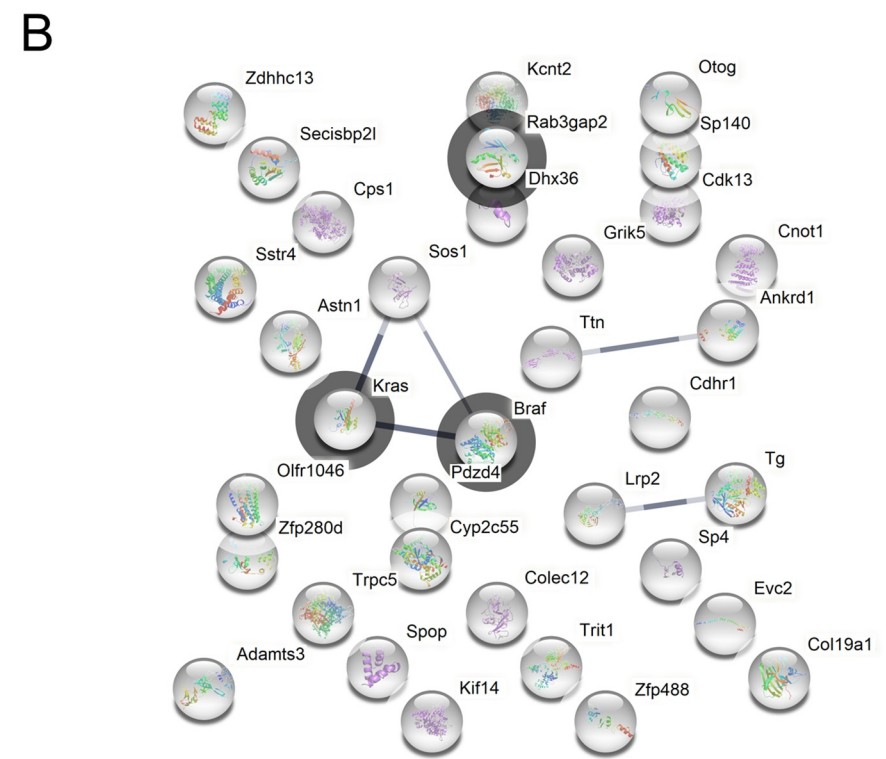

**Fig 1. Identification of non-canonical genomic driver mutations.** (**A**) Putative genomic driver mutations determined from WES of tumors, plotted based on their mutagenic optimality determined by MDS for urethane-treated normalized to PBS-treated mouse lung tissues versus for PBS-treated mouse lung tissues. Red highlight mutations occurring in at least one human cancer patient. Thick circles denote mutants validated by Sanger sequencing. (**B**) String analysis of proteins with a putative genomic driver mutation identified from urethane-induced mouse lung tumors. Limited to those with mutations with a high VAF at amino acid positions that are mutated in at least one human cancer patient in genes with single mutation observed in at least five human cancer patients. Dark circle highlights mutations occurring in multiple mouse lung tumors.

'urethane—PBS' in **Figs 1A** and **3B**. For the mutation spectrum of WES, SNVs in all tumors were annotated by the 96 possible trinucleotide context substitutions and summed in each tumor. These counts were then converted to per tumor proportions and the average across all tumors were calculated and plotted.

### Mutations signature analysis of *KRAS*-mutant human lung cancer patients

Contribution to each mutation class by each mutation signature was based on the file 'COSMIC_Mutational_Signatures_v3.1' downloadable from COSMIC (https://cancer.sanger.ac.uk/cosmic/signatures/SBS/index.tt) [27]. Contribution of mutation signature in each tumor were extracted from the files "PCAWG_sigProfiler _SBS _signatures_in_samples.csv" and "TCGA_-WES_sigProfiler_SBS_signatures_in_ samples.csv" under synapse ID syn11804040 at Synapse (https://www.synapse.org/). KRAS mutations detected in each tumor were extracted from through ICGC Data Portal [26] by cross-referencing the donor ID matching the sample ID in the files "PCAWG_sigProfiler_SBS_signatures_in_samples.csv" and "TCGA_WES_sigProfiler_ SBS_signatures_in_samples.csv" with the donor ID of KRAS-mutant LUADs. For each KRAS-mutant tumor, the contribution of individual signature to each substitution type in that tumor was calculated by *1*) converting the mutation counts from each signature to per tumor proportions; *2*) multiplying the calculated per tumor proportions by the percentage of contribution to different substitutions by each signature recorded in the file 'COSMIC_Mutational_-Signatures _v3.1'; *3*) normalizing the calculated value so the sum of the values from all mutation signatures for each substitution in a tumor is 1. The calculated contribution of individual signature to the substitution matching the *KRAS* mutation in that tumor were plotted in **Fig 5B**. The signature mismatch score in **Fig 5B** was calculated as the contribution to the substitution matching *KRAS* mutation in that tumor minus the contribution to that tumor overall by the signature contributing the most to the substitution matching the *KRAS* mutation in that tumor. To calculate the normalized mutation burden for different substitutions in **Fig 5A**, the mutation counts from individual signatures was first estimated by multiplying the mutation counts for each signature in that tumor by the contribution of individual signature to each substitution type in that tumor. Counts from all signatures in that tumor were then summed for each substitution type and converted to per tumor proportions of the counts for all substitutions. The per tumor proportion for the substitution matching the *KRAS* mutation in that tumor was used as concordance score.

## Results

### Identification of non-canonical genomic driver mutations

To identify non-canonical genomic driver mutations of urethane carcinogenesis, we performed WES analysis on a panel of 23 tumors derived from urethane-exposed mice with none, one, or two of the aforementioned $Kras^{ex3op}$ alleles upon recombination of the two $Trp53^{fl}$ alleles specifically in the lung (**S1 Table**), as loss of p53 yields tumors with either canonical $Q_{61}L/R$ or non-canonical $G_{12}D$ *Kras* mutations in these backgrounds [11]. The resulting

datasets were then screened for nonsynonymous SNV in any gene with a *V*ariant *A*llele *Fr*equency (VAF) within three standard deviations of the mean of that detected for all *Kras* mutations, suggestive of a truncal mutation (**S2 Table**). These were then censored for having at least one mutation detected in the human counterpart from at least five human cancer patients in ICGC data portal [26], suggestive of a tumor-related gene (**Fig 1A**). Following this, candidate genes were screened for mutations at the same codon in human tumors, suggestive of a genomic driver mutation (**S2 Table**). This revealed 0 to 5 putative genomic driver mutations per tumor, so we further defined a putative initiating genomic driver mutation as having the identical mutation in more than ten human cancer patients. As expected, this analysis identified the substitutions giving rise to the canonical $Q_{61}L$ and $Q_{61}R$ mutations in *Kras*, as well as three other non-canonical mutations, the previously described $G_{12}D$ mutation in *Kras*, a novel $Q_{61}H$ mutation in *Kras*, and finally, a novel Braf$^{V637E}$ mutation. Outside of these five mutations, a putative initiating genomic driver mutation could not be identified in the remaining tumors by these criteria. Furthermore, STRING analysis of all identified mutant proteins did not identify any obvious relationship beyond the MAPK pathway (**Fig 1B**). Censoring for *any* mutation in a panel of 460 genes associated with human cancers [25] found a C*A*N ➡ C*T/G*N mutation with a high VAF in *Ctnnb1* and in *Spen* in one Kras/Braf mutation-negative tumor each (**Fig 2**), but these mutations were not conserved in human cancers. Thus, we confirm the previous identification of the non-canonical $G_{12}D$ mutation in *Kras*, but now extend this to include another non-canonical *Kras* mutation, $Q_{61}H$, and finally, identify Braf$^{V637E}$ as a potential new initiating genomic driver mutation in urethane-induced tumors, thereby expanding the mutational bias of this carcinogen to yet another oncogene activating the MAPK pathway.

## Two types of non-canonical genomic driver mutations

To explore the relationship of the non-canonical mutations with urethane mutagenesis, we plotted the frequency of all possible SNVs determined by WES analysis of tumors, which we term the 'WES mutation signature' for ease of discussion. To determine the urethane mutation signature, we similarly plotted all SNVs from previous generated MDS sequencing data [7, 11] of *Kras* exon 1 and exon 2 as well as from *Hras* exon 2 from the lungs from mice exposed to urethane normalized to the same MDS sequencing analysis of the lungs of mice injected with the vehicle PBS (**S3 Table**), which we term the 'urethane-PBS mutation signature'. To parse out contributions by other possible mutagenic events in the urethane-PBS signature, we further we plotted the SNVs from the raw MDS sequencing data for just urethane, which we term the 'urethane mutation signature', and for just PBS, which we term the 'PBS mutation signature' (**Fig 3A**). We next compared the optimality of all putative genomic driver mutations detected in the tumors by WES analysis to the PBS versus urethane-PBS mutation signatures. As expected, the $Q_{61}L$ and to a lesser degree $Q_{61}R$ mutations in *Kras* were more similar to the urethane-PBS mutation signature. Similarly, the Braf$^{V637E}$ mutation was a better match to this signature as well. On the other hand, the $G_{12}D$ and $Q_{61}H$ mutations in *Kras* tracked with the PBS signature (**Figs 1A** and **3B**). Thus, we identified genomic driver mutations both matching and not matching the mutation signature of urethane.

## The non-canonical Braf$^{V637E}$ genomic driver mutation matches the mutation signature of urethane

As noted above, a Braf$^{V637E}$ mutation was identified in five tumors (**Figs 1A** and **3B**), which corresponds to the oncogenic BRAF$^{V600E}$ mutation [28] found in ~2% of human lung adenocarcinomas [26]. As a Braf$^{V637E}$ mutation had not previously been ascribed to urethane carcinogenesis, but has been detected in liver or lung tumors induced by other carcinogens or

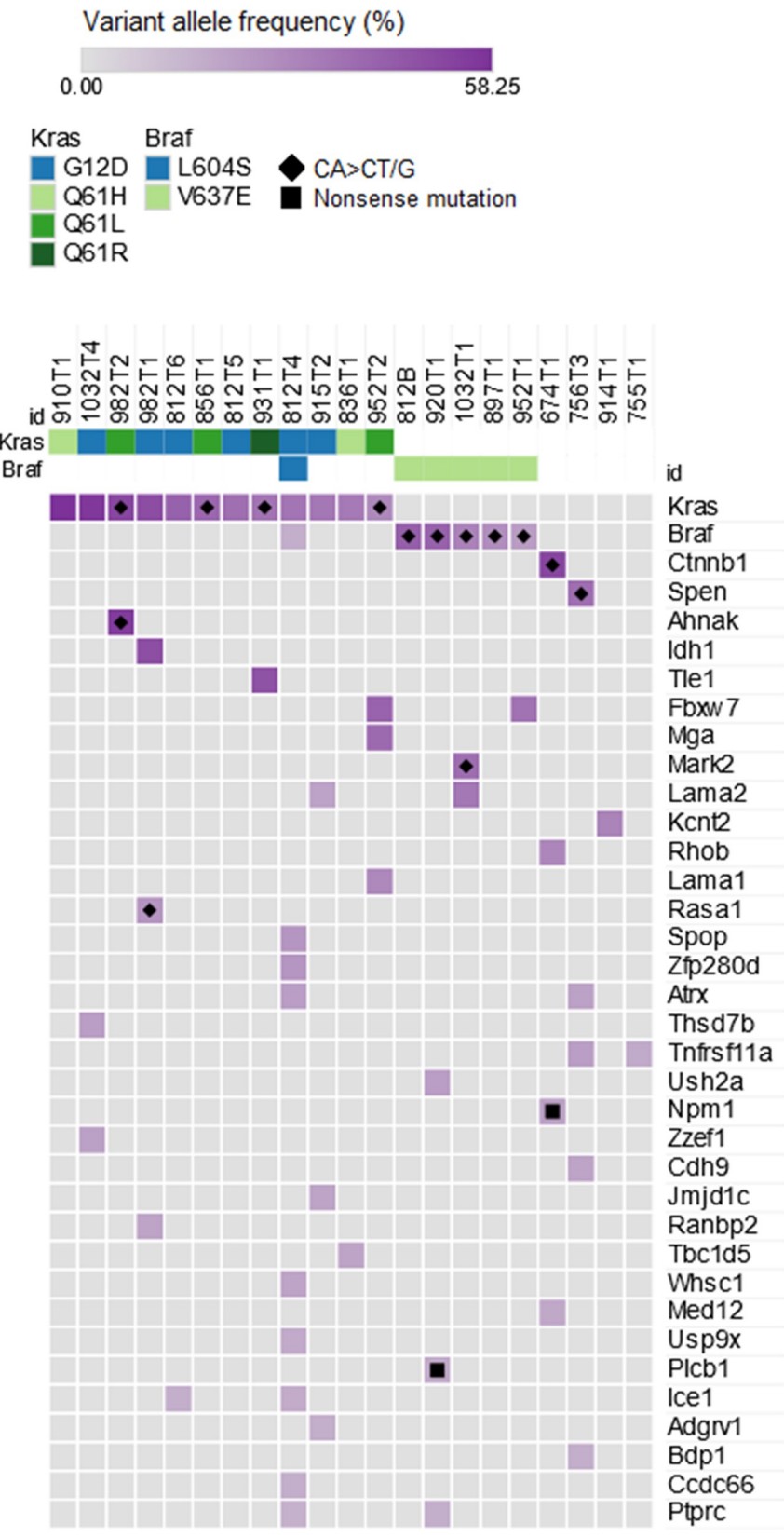

**Fig 2. Identification of non-canonical genomic driver mutations.** Putative genomic driver mutations (missense and nonsense mutations) in a list of 460 cancer genes identified in urethane-induced mouse lung tumors with variant allele frequency higher than 15.7% (calculated as the mean-3SD of the VAFs of *Kras* mutations). Two tumors with 0 putative genomic driver mutations identified by these criteria were not shown.

arising spontaneously in mice [29], we confirmed that this mutation was indeed somatic by Sanger sequencing exon 18 of *Braf* from the tumor and matched lung (normal) tissue (**S2 Table**), and further, not a product of p53 loss, as the same mutation was detected in urethane-induced tumors in a p53 wild-type background (**Fig 4A**). As noted above, the Braf$^{V637E}$ mutation has a greater overlap with the urethane-PBS mutation signature, which is even more evident upon plotting the ratio of the optimality of the urethane-PBS to PBS mutation signatures (**Figs 1A** and, **3B**). Indeed, the mutation giving rise to Braf$^{V637E}$ is G$_{1909}$*T*G➙G*A*G, which on the reverse strand matches the C*A*N➙C*T/G*N consensus sequence of urethane mutations determined by MDS after urethane exposure [7], although admittedly having a C at the 3' position makes it one of the less common mutations of this consensus (**Fig 1A**). Importantly, we did not detect V$_{637}$M, K$_{638}$E, G$_{506}$A, D$_{631}$N, or N$_{618}$S mutations, the murine equivalent to the human V$_{600}$M, K$_{601}$E, G$_{469}$A, D$_{594}$N, and N$_{581}$S oncogenic *BRAF* mutations [30, 31]. In agreement, none of these mutations contained the urethane mutation C*A*N➙C*T/G*N consensus, and related, these mutations were largely discordant with the urethane-PBS signature, and instead mostly tracked with the PBS signature (**Fig 4B**). Thus, akin to Kras$^{Q61L/R}$ mutations, the mutation specificity of the carcinogen appears to underlie the type of oncogenic *Braf* mutations promoting urethane carcinogenesis.

## The non-canonical *Kras*$^{G12D}$ and *Kras*$^{Q61H}$ mutations match the mutation signature of PBS

Interestingly, we find that the urethane mutation signature is more complex than the urethane-PBS mutation signature, with the former sharing more similarity with the WES mutation signature. Comparing the urethane and PBS mutation signatures revealed extensive overlap, suggesting that the origin of this increased complexity may arise from a mutation signature in normal cells (**Fig 3A**). One important caveat to this interpretation is a technical explanation for the SNVs detected in the PBS control tissue [32]. However, MDS was originally designed to detected the *de novo* mutations arising in bacteria populations [6], it is tempting to speculate that the incredible sensitivity of MDS may similarly be reporting on an endogenous mutagenic process in normal cells in mammals. Following, these SNVs would then become clonal in the tumors, adding to the complexity of the WES mutation signature and its similarity to the PBS mutation signature. As such, the discordance between the G$_{12}$D and Q$_{61}$H mutations in *Kras* with the urethane-PBS mutation signature, and concordance with the urethane, PBS, and WES mutation signatures, points to these mutations arising independent of the mutagenic activity of urethane, as suggested for other carcinogens [29]. Although admittedly rather speculative, one possibility is that these mutations were perhaps derived from the same process that gave rise to the PBS mutation signature.

## Mutagenic specificity and oncogenic selection influence the RAS mutation patterning in human lung adenocarcinoma

The above findings are consistent with the mutational bias of urethane generating Kras$^{Q61L/R}$ and Braf$^{V637E}$ mutations in derived tumors, but not the G$_{12}$D and Q$_{61}$H mutations. As urethane is still required to induce tumors with the latter mutations, the carcinogen may be acting in a promotional manner to foster the expansion of cells with non-canonical oncogenic

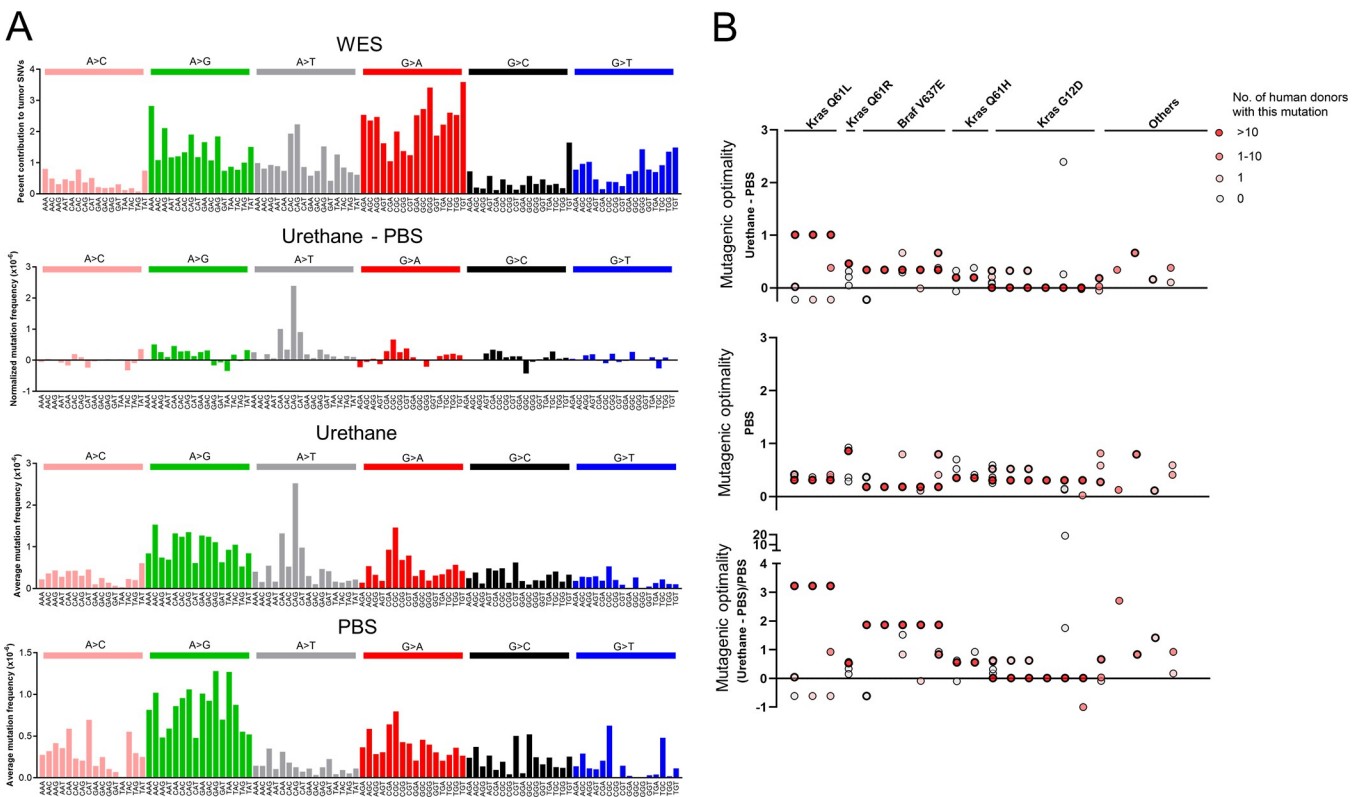

**Fig 3. Multiple mutational processes contribute to putative genomic driver mutations.** (**A**) Mutation spectrum measured by MDS for PBS, urethane, urethane normalized to PBS-treated mouse lung tissues and mutation spectrum measured by WES for urethane-induced mouse lung tumors. (**B**) Mutagenic optimality of putative genomic driver mutations identified by MDS for urethane normalized to PBS-treated and PBS-treated mouse lung tissues as well as the ratio of the former divided by the latter. Data are plotted for individual tumors and the genomic driver mutations occurring in more than 10 human cancer patients are indicated at the top. Others refers to tumors without a genomic driver mutation identified. Thick circles denote mutants validated by Sanger sequencing.

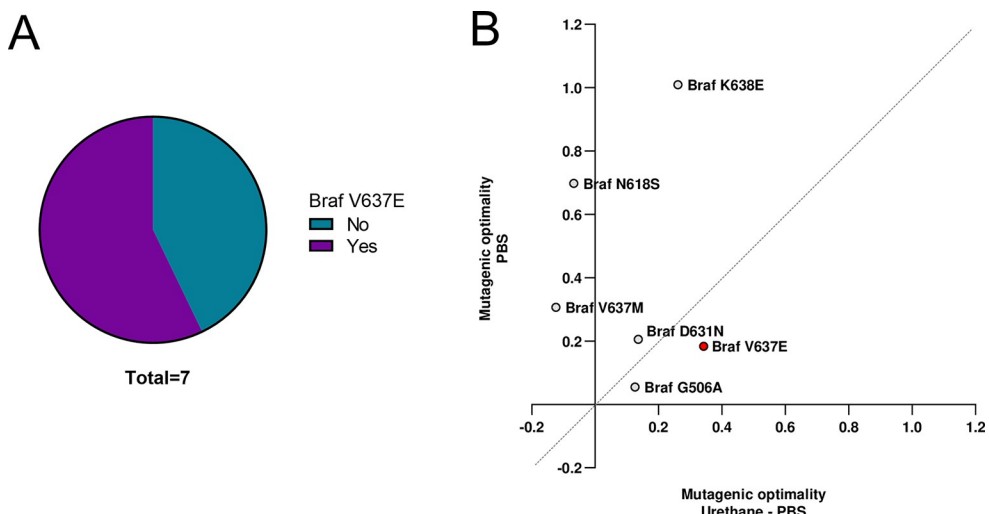

**Fig 4. Multiple mutational processes contribute to putative genomic driver mutations.** (**A**) Detection of $Braf^{V637E}$ mutation in urethane-induced p53 wildtype lung tumors using Sanger sequencing. (**B**) Mutagenic optimality of the mouse equivalent of human oncogenic BRAF mutations identified by MDS for urethane-treated normalized to PBS-treated mouse lung tissues versus for PBS-treated mouse lung tissues. Red highlight $Braf^{V637E}$ mutation detected in urethane-induced mouse lung tumors.

mutations. In support, recent _W_hole-_G_enome _Se_quencing (WGS) analysis of numerous different carcinogen-induced tumors found few tumors whereby the mutation signature of the carcinogen matched the genomic driver mutation [29]. Recent analysis of human cancer genomes also showed that in addition to mutational processes, functional selection based on tissue-of-origin, signaling property of _KRAS_ allele, as well as cooperating genetic events, also shape the mutational bias of _KRAS_ gene across cancer types [33]. Thus, these previous studies collectively argue that in some cases, carcinogens may function as non-mutagenic tumor promoting agent contributing to the selection of specific _Kras_ mutations in cancer. To determine if there is evidence for such a promotional activity in the selection of oncogenic _KRAS_ mutations in humans, we extracted the mutation signatures in the COSMIC database [27, 34] for human KRAS-mutant lung adenocarcinomas. We then compared the concordance of all missense _KRAS_ mutations extracted from ICGC data portal for lung adenocarcinomas [26] to the aforementioned mutation signatures (**Fig 5A**). Consistent with the mouse data, _KRAS_ mutations in human lung adenocarcinomas displayed different levels of concordance with the mutation spectrum predicted from the mutation signatures. In more detail, the $G_{12}C$ mutation in KRAS$^{G12C}$-mutant tumors had high concordance with the mutation spectrum of these tumors, consistent with the contribution of SBS4, associated with tobacco smoking, to $G_{12}C$ mutations [35]. KRAS$^{G12V}$-mutant tumors have a similar mutation spectrum, and the mutation spectrum of both KRAS$^{G12C}$- and KRAS$^{G12V}$-mutant tumors favors these mutations, suggesting that either mutation arise from a potentially common mutational event. However, the mutations spectrum of Kras$^{G12C/V}$-mutant human lung adenocarcinomas also strongly favor $G_{13}C$, $G_{13}V$, $Q_{61}K$, and $L_{19}F$ mutations, which are not recovered in this cancer. In fact, $G_{12}A$, and to a lesser degree $G_{12}D$ and other _KRAS_ mutations, are discordant with the mutation spectrum of the corresponding tumors. As in urethane carcinogenesis, we thus find evidence for secondary mutational process underlying some of these mutations. Namely, comparing the known _KRAS_ mutations to the different mutation signatures present in each tumor revealed that while most oncogenic _KRAS_ mutations tracked to the dominant mutation signature of each tumor, nearly a quarter of the KRAS$^{G12/13C}$ mutations tracked with a minor mutation signature (**Fig 5B**). Thus, the concordance of the mutation signature with the genomic driver mutation supports a mutational event leading to a tumor, while discordance between these two suggests another possible mode of tumor initiation.

## Discussion

Urethane displays a bias towards C_A_N➡C_T/G_N mutations that perfectly match with the Kras$^{Q61L/R}$ genomic driver mutations detected in developing tumors. We also show that the same holds true for the Braf$^{V637E}$ oncogenic mutation, which again may explain its origins. Why the latter occurs less often remains to be resolved, but the two oncoproteins are not fungible, as Kras acts upstream of Braf and engages different proteins. Additionally, the site for a Braf$^{V637E}$ mutation diverges from the most idea mutation consensus sequence more than the Kras$^{Q61L/R}$ mutation. Nevertheless, as mentioned, these mutations appear to be a product of the mutational bias of the carcinogen.

The same cannot be said for the non-canonical mutations detected in _Kras_, which raises the question as to how they arose in the first place. One hint to their origins comes from the finding that the G➡A substitutions responsible for these mutations were detected by MDS in the PBS-treated cohort. While we cannot rule out that these mutations are induced by urethane at a level beyond the detection limit of MDS, a very important consideration, these data do suggest the intriguing possibility that the point mutations encoding $G_{12}D$ and $Q_{61}H$ in _Kras_ may instead be urethane-independent. Perhaps, and again rather speculatively, these mutations and

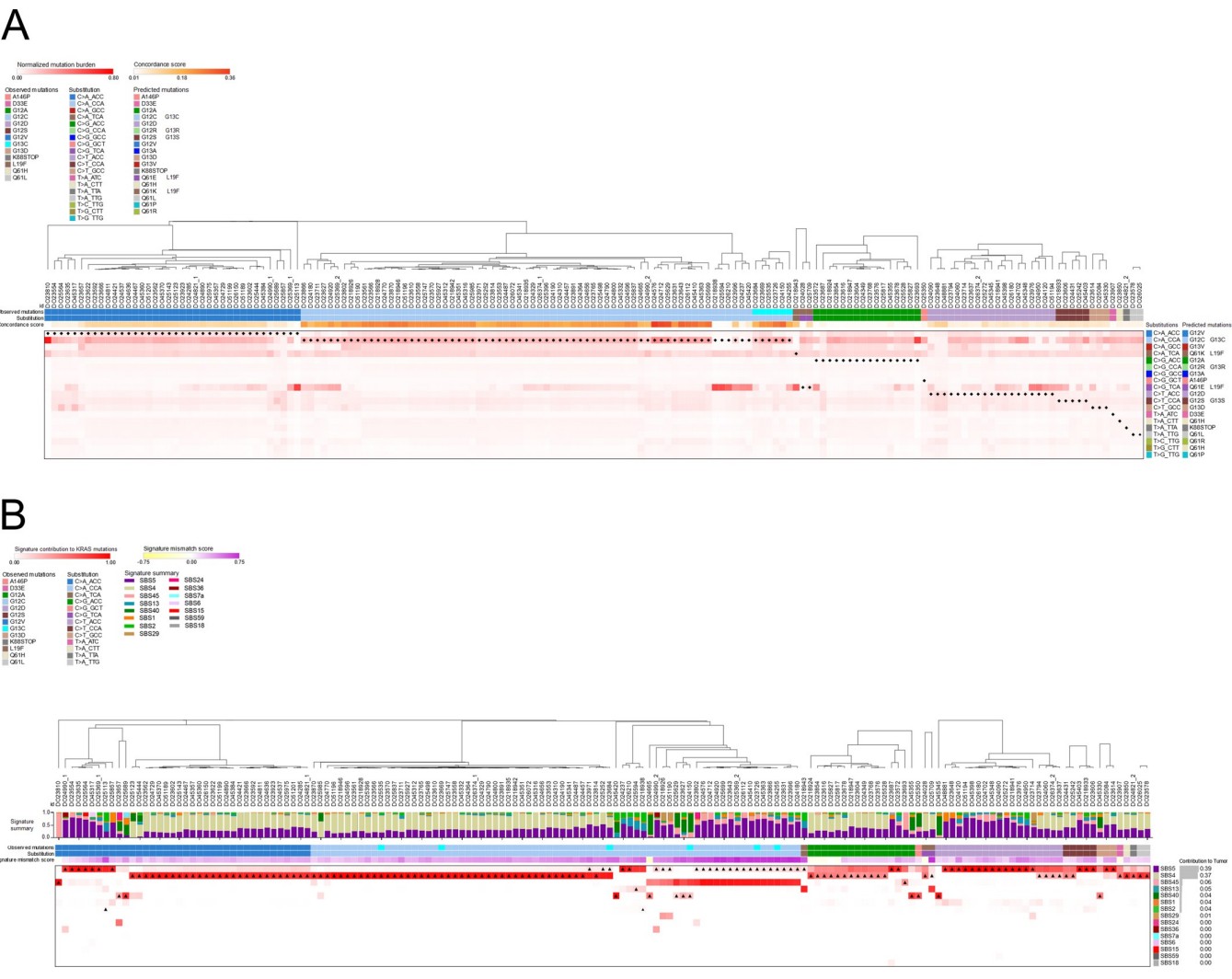

**Fig 5. Multifactorial processes determine the type of *KRAS* mutations observed in human lung adenocarcinomas.** (**A**) Heatmap of the mutation burden for substitutions matching *KRAS* oncogenic mutations in human lung adenocarcinomas estimated from the mutation signatures active in these tumors and normalized to per tumor proportions. Black diamond indicates the substitution type matching the *KRAS* mutation in that tumor. The *KRAS* mutation and the substitution type for the *KRAS* mutation for each tumor were shown below the donors. Concordance score reflects the degree of concordance between the most likely *KRAS* mutation predicted by the estimated mutation burden and the actual *KRAS* mutation observed in that tumor. (**B**) Heatmap of the contribution of mutation signatures to *KRAS* mutations detected in human lung adenocarcinomas. Black triangles indicate the dominant mutation signature in that tumor measured by contribution to total mutation burden. Signature summary displays the contribution of mutation signatures to total mutation burden in each tumor. The *KRAS* mutation and the substitution type for the *KRAS* mutation for each tumor were shown below the signature summary. Signature mismatch score reflects the degree of the discordance between the contribution to mutations overall and the contribution to *KRAS* mutation alone for the mutation signature most responsible the *KRAS* mutation observed in that tumor.

SNVs detected by ultra-sensitive MDS sequencing of the lungs of control mice injected with the vehicle PBS are generated by the same process. As urethane is still required for tumorigenesis, perhaps the carcinogen causes another mutation that favors the expansion of a pre-existing or subsequently induced oncogenic Kras$^{G12D/Q61H}$ mutation due to an endogenous mutagenic process. In support, oncogenic mutations have been detected in normal human tissues without evidence of cancer [36]. Moreover, Kras$^{G12D}$-mutant urethane-induced tumors had an average of 14 other truncal (high VAF) C__A__➡C__T/G__ mutations, including the recurrent mutations Zfp991$^{H211R}$, Nccrp1$^{M197T}$, and Tnn$^{V863A}$. These tumors had even more mutant genes when lowering the threshold to the more general urethane consensus A➡T/G, which identified the

cancer-related mutations Rab3gap2$^{Y467C}$, Zfp280d$^{N593S}$, Cps1$^{F565I}$, and Evc2$^{Q510L}$ as well as the recurrent mutations Gucy1a3$^{E232V}$, Ice1$^{E1919V}$, Tcp10c$^{I194T}$, Vmn1r82$^{I47V}$, and Flna$^{S2276T}$ (**S2 Table**). Alternatively, urethane may simply have non-mutagenic activities [37, 38] that promote the expansion of pre-existing Kras$^{G12D/Q61H}$-mutant cells, consistent with recent finding of a common discordance between genomic driver mutations and carcinogen-specific mutation signatures in a wide spectrum of carcinogen-induced tumors [29]. Collectively, these findings suggest that selection, rather than mutational specificity, may be the dominating factor underlying which RAS mutation drives tumor initiation.

We also identified four tumors that lacked an oncogenic mutation in either *Kras* or *Braf*. Two of these tumors had a C*A*N➡C*T/G*N mutation with a high VAF in either *Ctnnb1* or *Spen*, and neither was accompanied by a well-characterized oncogenic mutation, suggestive of a genomic driver mutation. *Ctnnb1* encodes β-catenin, which is well known to play a role in cancer [39], and this gene is mutated in other carcinogen as well as spontaneous murine tumors [29]. Furthermore, the detected mutation, S$_{23}$C, occurs at a site of phosphorylation [40] and O-GlcNAcylation [41]. However, while an S$_{23}$R mutation has been reported in human cancers [42], this mutant protein does not transform cells [40]. Furthermore, the specific mutation we detected was S23C, not S23R, and an S23C mutation has not been reported in human cancer, and hence failed our last criteria for being classified as a genomic driver mutation. *Spen* encodes a transcriptional repressor of Notch signaling, again a pathway implicated in cancer [43]. While this gene is found mutated at a low frequency in a variety of human cancers [44–46], the identified mutation Q$_{179}$L has not been reported, and hence also failed the last criteria for being classified as a genomic driver mutation in urethane-induced tumors. Thus, the nature of the genomic driver mutation in these two tumors remains unclear. Similarly, we were unable to assign an obvious genomic driver mutation to the other two tumors. However, activation of MAPK pathway may yet still play a role in the progression of tumors negative for *Kras/Braf* genomic driver mutations. Specifically, copy number analysis revealed a gain of chromosome 6, which contains both the *Kras* and the *Braf* gene, in some of these tumors (**S1 Fig**). Furthermore, copy number analysis also detected the loss of chromosome 9 in these tumors (**S1 Fig**), which the *Keap1* and *Setd2* genes, both of which have been shown to function as tumor suppressors in *Kras$^{G12D}$*-driven mouse model of lung adenocarcinoma [47, 48]. Thus, there are provocative candidate genomic driver mutations to explore in these other tumors.

As noted, we uncovered two types of genomic driver mutations promoting urethane carcinogenesis, those that matched the mutation signature of the carcinogen, and those that did not. It is worth noting that discordance, rather than concordance of a genomic driver mutation with the tumor mutation signature is indeed observed in human cancers. Case in point, the mutation giving rise to oncogenic *BRAF*$^{V600E}$ fails to match the mutation signature of UV light exposure [35]. The same can be said for *BRAF*$^{V600E}$ in colorectal cancer, *PTEN*$^{R130G}$ in uterine carcinoma [35], *TP53* mutations in APOBEC+ human cancers [49], and a number of other genomic driver mutations in melanoma [25]. In agreement with a previous, and far more exhaustive analysis of human tumor genomic data [33], we find that the majority of the human lung adenocarcinomas analyzed had a mutation spectrum favoring a subset of mutations that included the known oncogenic *KRAS* mutation of the tumor. Nevertheless, a minority of these tumors exhibited a discordance between the genomic driver mutation and the mutation spectrum. In some cases this could be attributed to a mutagenic spectrum that was either extremely diverse (*e.g.* tracking with many RAS mutations) or specific (*e.g.* tracking with no RAS mutations), but there were definitely cases of discordance with a mutagenic signature normally tracking with a specific *KRAS* mutation. Such a discordance points to biological selection as a factor in the establishment of the genomic driver mutation in these cancers.

Interestingly, there was no mutagenic signature specific for just one type of oncogenic *KRAS* mutation. Whether this reflects mutagenic processes that give rise to the mutation signatures in human lung adenocarcinomas typically being less specific than urethane in mice, the mutagenic signature at tumor initiation being degraded over time due to additional mutagenic processes, or a frank inability of a highly specific mutagenic process to initiate lung tumorigenesis remains to be determined. Nevertheless, we hypothesize that the RAS mutation patterns of human lung adenocarcinomas could be ascribed, at least in part, to varying degrees of mutagenic specificity, akin to how we envision urethane inducing tumors with a Kras$^{Q61L/R}$ or Braf$^{V637E}$ genomic driver mutation, and biological selection, akin to how we speculate that urethane induces tumors with a Kras$^{G12D/Q61H}$ genomic driver mutation.

## Supporting information

**S1 Fig. Copy number analysis reveals copy number changes in chromosome 6 and 9 in multiple tumors.** (**A**) Heatmap of log$_2$-transformed copy number ratios showing areas of genomic gain or loss within each tumor. Columns correspond to genomic bins and rows correspond to individual tumors. Tumors are annotated by the type of Kras or Braf mutations present in the tumor. (**B**) Cancer-associated genes with copy number gain (log$_2$ ratio > 0.25) or loss (log$_2$ ratio < -0.25) in chromosome 6 and 9 in each tumor. Included genes are from the list of 460 cancer genes examined in Fig 2.
(TIF)

**S1 Table. Information of sequenced tumors.**
(XLSX)

**S2 Table. SNVs and putative genomic driver mutations detected in sequenced tumors.**
(XLSX)

**S3 Table. Mutation spectrum of MDS.**
(XLSX)

**S4 Table. Sanger sequencing primers.**
(XLSX)

**S1 File. Guideline checklist.**
(PDF)

## Acknowledgments

We thank Dr. David MacAlpine (Duke University) for thoughtful discussions and colleagues from Duke Cancer Institute Sequencing and Genomic Technologies Shared Resource for preparing the sequencing library for whole exome sequencing and performing the sequencing of next generation sequencing libraries.

## Author Contributions

**Conceptualization:** Siqi Li, Christopher M. Counter.

**Data curation:** Siqi Li.

**Formal analysis:** Siqi Li.

**Funding acquisition:** Christopher M. Counter.

**Investigation:** Siqi Li.

**Methodology:** Siqi Li.

**Project administration:** Christopher M. Counter.

**Resources:** Christopher M. Counter.

**Supervision:** Christopher M. Counter.

**Validation:** Siqi Li.

**Visualization:** Siqi Li.

**Writing – original draft:** Siqi Li, Christopher M. Counter.

**Writing – review & editing:** Siqi Li, Christopher M. Counter.

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
