## [Decision Letter · Decision Letter 0]

21 Jan 2022

PONE-D-21-38368Non-canonical driver mutations of urethane carcinogenesisPLOS ONE

Dear Dr. Counter,

Thank you for submitting your manuscript to PLOS ONE. After careful consideration, we feel that it has merit but does not fully meet PLOS ONE’s publication criteria as it currently stands. Therefore, we invite you to submit a revised version of the manuscript that addresses the points raised during the review process.

I would like to sincerely apologize for the long time it has taken to complete the review process. Even though you submitted this manuscript well in early December, the holidays were approaching, and many qualified Reviewers declined to review the manuscript. Two accepted the invitation to review, but one was automatically uninvited by the PLOS ONE review mechanism shortly after the New Year. This is an unfortunate uninvited process, and I have let the journal editors know that there are times when it is not helpful. I was not able to get this Reviewer to provide a review. However, there is one reasonable review from a person knowledgeable about the topic of your manuscript. Kindly address the concerns of this Reviewer.

Please submit your revised manuscript Mar 07 2022 11:59PM. If you will need more time than this to complete your revisions, please reply to this message or contact the journal office at plosone@plos.org. Please include the following items when submitting your revised manuscript:A rebuttal letter that responds to each point raised by the academic editor and reviewer(s). You should upload this letter as a separate file labeled 'Response to Reviewers'.A marked-up copy of your manuscript that highlights changes made to the original version. You should upload this as a separate file labeled 'Revised Manuscript with Track Changes'.An unmarked version of your revised paper without tracked changes. You should upload this as a separate file labeled 'Manuscript'.If applicable, we recommend that you deposit your laboratory protocols in protocols.io to enhance the reproducibility of your results. Protocols.io assigns your protocol its own identifier (DOI) so that it can be cited independently in the future. For instructions see: https://journals.plos.org/plosone/s/submission-guidelines#loc-laboratory-protocols. Additionally, PLOS ONE offers an option for publishing peer-reviewed Lab Protocol articles, which describe protocols hosted on protocols.io. Read more information on sharing protocols at https://plos.org/protocols?utm_medium=editorial-email&utm_source=authorletters&utm_campaign=protocols.

We look forward to receiving your revised manuscript.

Kind regards,

Shawn Ahmed

Academic Editor

PLOS ONE

Journal Requirements:

2. To comply with PLOS ONE submissions requirements, in your Methods section, please provide additional information on the animal research and ensure you have included details on (1) methods of sacrifice, (2) methods of anesthesia and/or analgesia, (3) efforts to alleviate suffering, (4) basic housing and breeding, (5) health monitoring.

3. When the authors resubmit, please check to ensure they have included a completed ARRIVE checklist as a Supporting Information file (and send back if they have not). Do not ping for follow-up.

"As part of your revision, please complete and submit a copy of the Full ARRIVE 2.0 Guidelines checklist, a document that aims to improve experimental reporting and reproducibility of animal studies for purposes of post-publication data analysis and reproducibility: https://arriveguidelines.org/sites/arrive/files/Author%20Checklist%20-%20Full.pdf (PDF). Please include your completed checklist as a Supporting Information file. Note that if your paper is accepted for publication, this checklist will be published as part of your article.

(This work was supported by the National Cancer Institute (USA) grants R01CA123031 and P01CA203657 to CMC.  The funders did not play any role in the study design, data collection and analysis, decision to publish, or preparation of the manuscript.)

(CMC is co-Founder of Merlon Inc, which did not play any role in the study design, data collection and analysis, decision to publish, or preparation of the manuscript.)

We note that one or more of the authors are employed by a commercial company: name of commercial company. 

6. Please update your submission to use the PLOS LaTeX template. The template and more information on our requirements for LaTeX submissions can be found at http://journals.plos.org/plosone/s/latex.

7. Thank you for stating the following in the Acknowledgments Section of your manuscript: 

(We thank Dr. David MacAlpine (Duke University) for thoughtful discussions, and colleagues from Duke Cancer Institute Sequencing and Genomic Technologies Shared Resource for preparing the sequencing library for whole exome sequencing and performing the sequencing of next generation sequencing libraries, a shared resource supported by P30CA014236 of the Duke Cancer Institute. This work was supported by the National Cancer Institute (R01CA123031 and P01CA203657 to CMC))

(This work was supported by the National Cancer Institute (USA) grants R01CA123031 and P01CA203657 to CMC.  The funders did not play any role in the study design, data collection and analysis, decision to publish, or preparation of the manuscript.)

8. Please include your full ethics statement in the ‘Methods’ section of your manuscript file. In your statement, please include the full name of the IRB or ethics committee who approved or waived your study, as well as whether or not you obtained informed written or verbal consent. If consent was waived for your study, please include this information in your statement as well. 

Reviewers' comments:

Reviewer's Responses to Questions

**Comments to the Author**

1. Is the manuscript technically sound, and do the data support the conclusions?

Reviewer #1: Yes

2. Has the statistical analysis been performed appropriately and rigorously? 

Reviewer #1: Yes

3. Have the authors made all data underlying the findings in their manuscript fully available?

Reviewer #1: Yes

4. Is the manuscript presented in an intelligible fashion and written in standard English?

Reviewer #1: Yes

5. Review Comments to the Author

Reviewer #1: Thank you for the opportunity to review the manuscript entitled "Non-canonical driver mutations of urethane carcinogenesis" by Drs. Li and Counter for PLOS ONE. This is an important manuscript, describing mutations observed in pulmonary tumors that arise in mice as a result of urethane exposure when the typical mutation observed in this scenario (Kras Q61L/R) is selected against. This is an important study in terms of describing fully the mechanism of urethane carcinogenesis as well as defining pathways that can functionally substitute for Kras Q61L. Overall the manuscript is well written and the experiments are well done. I have several suggestions for further analyses that could strengthen the conclusions of the paper. These suggestions and some additional comments are outlined below

1) Please be careful about the use of the term "driver" without functional validation. The functional validation, such as the possibility of creating transgenic mouse models expressing Rab3gap2 Y467C and KrasG12D, is so exciting, but clearly outside the scope of the current paper. Perhaps the term "genomic driver" would be more accurate.

2) The authors could perform copy number analysis (CNVkit) using the whole exome sequencing data generated for this manuscript. This analysis could provide possible genomic drivers for the tumors that don't have an obvious SNV genomic driver.

3) Have the authors considered performing subclonal analysis and/or attempting to predict the timing of the genetic events contributing to carcinogenesis in these tumors? This analysis could add depth to the description of urethane carcinogenesis. Please see PLoS Genet

. 2015 Mar 13;11(3):e1005075. doi: 10.1371/journal.pgen.1005075. eCollection 2015 Mar. for an example of this analysis. WES of the germline in these mice would be required to conduct the analysis

6. PLOS authors have the option to publish the peer review history of their article (what does this mean?). If published, this will include your full peer review and any attached files.

Reviewer #1: No

---

## [Author Response · Author response to Decision Letter 0]

6 Mar 2022

see attached files "Cover letter" and "Response to reviewer"

---

## [Decision Letter · Decision Letter 1]

4 Apr 2022

Non-canonical genomic driver mutations of urethane carcinogenesis

PONE-D-21-38368R1

Dear Dr. Counter,

Congratulations on an interesting manuscript that hypothesizes an interaction between carcinogen-induced and spontaneous mutations in the context of tumor development.  I am pleased to inform you that your manuscript has been judged scientifically suitable for publication and will be formally accepted for publication once it meets all outstanding technical requirements.

Kind regards,

Shawn Ahmed

Academic Editor

PLOS ONE

ps. How about them Tar Heels? This might be a fitting ending to a storied career with many titles to reflect on.

Additional Editor Comments (optional):

Your manuscript looks pretty clean, but I offer a few minor comments:

Line 30: ‘for specific genomic driver mutation’. Mutations?

Line 30: ‘Small Nucleotide Variations (SNVs)’. I am not familiar with this term. There is a related term ‘Single Nucleotide Variations (SNVs) that is more prominent in the literature. If you choose to use Small Nucleotide Variations, perhaps provide a reference that helps to clarify how this differs from Single Nucleotide Variations.

Line 33: ‘oncogenic mutation in the proto-oncogene BRAF conforming to the mutation signature’. Do you mean ‘oncogenic mutation in the proto-oncogene BRAF that conforms to the mutation signature’?

Line 162: ‘The previously published study has shown’. Perhaps ‘A previously published study has shown’

Line 164: ‘MDS has high mutation frequency’. Do you mean ‘MDS has a high mutation frequency’?

Line 191: ‘convert the mutation’ might be ‘converting the mutation’

Line 191: ‘multiply the calculated’ might be ‘multiplying the calculated’

Figure 2. What is the significance of the black squares of the nonsense mutations that matches the color of the urethane-induced mutations? Do the nonsense mutations also possess a urethane signature?

Figure 2. Could some of the purple mutations represent an unknown mutational signature of urethane? This might be apparent if there is a specific pattern of base change that is common to ⅓ or more of these mutations.

Line 422: ‘function as tumor suppressor in krasG12D-driven mouse model’. ‘function as tumor suppressors in a krasG12D-driven mouse model’

Line 440: ‘Interestingly, there was no mutagenic signature specific for just one KRAS mutation.’ Perhaps clarify what you mean by just one KRAS mutation? Does this refer to a single base change in KRAS? Does this mean that although urethane-induced mutations were apparent in KRAS, that there were also other mutations whose origin is unclear?

Reviewers' comments:

Reviewer's Responses to Questions

**Comments to the Author**

1. If the authors have adequately addressed your comments raised in a previous round of review and you feel that this manuscript is now acceptable for publication, you may indicate that here to bypass the “Comments to the Author” section, enter your conflict of interest statement in the “Confidential to Editor” section, and submit your "Accept" recommendation.

Reviewer #1: All comments have been addressed

2. Is the manuscript technically sound, and do the data support the conclusions?

Reviewer #1: Yes

3. Has the statistical analysis been performed appropriately and rigorously? 

Reviewer #1: Yes

4. Have the authors made all data underlying the findings in their manuscript fully available?

Reviewer #1: Yes

5. Is the manuscript presented in an intelligible fashion and written in standard English?

Reviewer #1: Yes

6. Review Comments to the Author

Reviewer #1: Thank you so much for your careful responses to my comments and I look forward to reading follow up papers to this important work.

7. PLOS authors have the option to publish the peer review history of their article (what does this mean?). If published, this will include your full peer review and any attached files.

Reviewer #1: No

---

## [Editor Report · Acceptance letter]

20 Apr 2022

PONE-D-21-38368R1 

Non-canonical genomic driver mutations of urethane carcinogenesis 

Dear Dr. Counter:

I'm pleased to inform you that your manuscript has been deemed suitable for publication in PLOS ONE. Congratulations! Your manuscript is now with our production department. 

Kind regards, 

on behalf of

Dr. Shawn Ahmed 

Academic Editor

PLOS ONE